# Effect of Potassium–Magnesium Sulfate on Intestinal Dissociation and Absorption Rate, Immune Function, and Expression of NLRP3 Inflammasome, Aquaporins and Ion Channels in Weaned Piglets

**DOI:** 10.3390/ani15121751

**Published:** 2025-06-13

**Authors:** Cui Zhu, Kaiyong Huang, Xiaolu Wen, Kaiguo Gao, Xuefen Yang, Zongyong Jiang, Shuting Cao, Li Wang

**Affiliations:** 1School of Animal Science and Technology, Foshan University, Foshan 528225, China; zhucui@fosu.edu.cn (C.Z.); huangky_300@kingkeyzn.com (K.H.); 2State Key Laboratory of Livestock and Poultry Breeding, Key Laboratory of Animal Nutrition and Feed Science in South China, Ministry of Agriculture and Rural Affairs, Guangdong Provincial Key Laboratory of Animal Breeding and Nutrition, Institute of Animal Science, Guangdong Academy of Agricultural Sciences, Guangzhou 510640, China; wenxiaolu@gdaas.cn (X.W.); gaokaiguo@gdaas.cn (K.G.); yangxuefen@gdaas.cn (X.Y.); jiangzy@gdaas.cn (Z.J.)

**Keywords:** magnesium potassium sulfate, weaned piglets, NLPR3, aquaporins, ion channel

## Abstract

Common ingredients in corn-soybean meal diets can provide sufficient magnesium and potassium for piglets. However, additional dietary magnesium and potassium supplements have become essential to support the higher performance and health requirements of piglets in modern pig production, especially during the weaning period. The current study aimed to investigate the influence of potassium–magnesium sulfate (PMS) on intestinal dissociation and absorption, immune function, and expression of the NLRP3 inflammasome, aquaporins, and ion channels in weaned piglets. These results illustrated that PMS supplementation could improve immune function and modulate intestinal genes involved in water and ion homeostasis to control post-weaning diarrhea in piglets.

## 1. Introduction

Within modern intensive swine farming systems, piglets are frequently subjected to various nutritional, environmental, and psychological stressors during the crucial transition period of weaning. These stressors can result in intestinal dysfunction, compromised immune function, diarrhea, growth retardation, and even mortality in weaned piglets [1]. To address this challenge, identifying nutritional approaches that mitigate weaning-associated stress and ensure optimal development of weaned piglets is critical [2].

Minerals are crucial nutrients that play a significant role in promoting metabolism and immune functions in animals. Both deficiency and excess minerals can adversely affect normal physiological functions and the overall health of animals [3,4]. Potassium and magnesium are essential minerals for maintaining normal growth and development, preventing disease, and ensuring the overall health of both humans and animals [5,6,7,8]. It is widely acknowledged that the concentrations of potassium and magnesium in conventional feed ingredients are generally adequate to fulfill the nutritional requirements of animals [4]. The swine nutrient requirement of the National Research Council (NRC) recommends that pigs receive 0.04% (400 mg/kg) Mg and 0.30–0.17% K per kg of diet on a dry matter (DM) basis [9]. However, due to the significant advancements in modern livestock production over the past few decades, the inclusion of additional magnesium and potassium supplements in the diet has become imperative to support higher animal performance and health requirements [4,10,11]. For example, dietary supplementation with magnesium has been demonstrated to enhance the quality of pork [12] and improve reproductive performance [13,14]. Furthermore, the inclusion of dietary potassium in feed or drinking water has been found to improve broiler performance under heat-stress conditions [15,16]. It is noteworthy that most of the aforementioned studies focused on the individual forms of magnesium or potassium in livestock production, with limited reports addressing the combined form of potassium and magnesium in weaned piglets.

Potassium magnesium sulfate (PMS) is a novel feed additive composed of natural compound salts that can simultaneously supply both potassium and magnesium to animals. Prior research has demonstrated that maternal dietary inclusion of PMS at 0.45% significantly elevated plasma levels of insulin-like growth factor 1 (IGF-1) and immunoglobulin A (IgA) in the colostrum while concurrently reducing plasma levels of TNF-α in sows and the incidence of intrauterine growth restriction (IUGR) in piglets [17]. Furthermore, our previous study indicated that dietary supplementation with PMS could enhance growth performance, decrease the incidence of post-weaning diarrhea, improve antioxidant capacity and intestinal immunity, and modulate the composition of the intestinal microbiota in weaned piglets [18]. However, the potential role of PMS in mitigating weaning stress in piglets through the activation of the NLRP3 inflammasome and intestinal expression of water and ion transporters remains unclear.

NLRP3 inflammasome is a crucial component of the innate immune system and plays a significant role in the regulation of inflammation and immune responses [19]. Its activation, triggered by microbial infections, endogenous danger signals, and environmental stimuli, frequently results in the release of pro-inflammatory cytokines, such as IL-1β and IL-18, which can induce pyroptosis and contribute to intestinal inflammation [20]. Moreover, increasing evidence has indicated that intracellular ion fluxes, including K^+^, Mg^2+^, Ca^2+^, and Cl^−^, are involved in regulating NLRP3 inflammasome activation [21,22,23]. As the NLRP3 inflammasome is involved in various inflammatory conditions, its modulation presents considerable therapeutic potential in health and disease [24,25]. Prior studies have demonstrated that dietary nutritional interventions could target the NLRP3 inflammasome to improve the intestinal health of weaned piglets [26,27,28]. Aquaporins (AQPs) and ion channels are integral membrane proteins that facilitate the transport of water and ions across cell membranes [29,30]. At least thirteen subtypes of AQPs (AQP0-12) have been identified in the gastrointestinal tract of mammals [31]. Numerous studies have shown that AQPs expression is closely related to diarrhea and intestinal inflammation [32,33], and that AQPs are important targets for dietary nutrients to regulate gut health [34]. Alterations in AQPs and ion channels serve as important biomarkers for maintaining intestinal fluid balance and electrolyte homeostasis in piglets [35]. Therefore, understanding how dietary PMS treatment regulates the NLRP3 inflammasome, AQPs, and ion channels in the intestine is important for the rational application of PMS in pig production.

Hence, this study aimed to investigate the dissociation and absorption of PMS in the gastrointestinal tract of weaned piglets, as well as to assess the effects of dietary PMS supplementation on serum immune function and intestinal expression of NLRP3, AQPs, and ion channels in weaned piglets.

## 2. Materials and Methods

### 2.1. Experiment 1

#### 2.1.1. In Vitro Dissociation Assay

The in vitro dissociation assay of PMS was conducted using an artificial gastric juice (pH 1.2) as previously described [36], which consisted of 3.2 g of gastric protease, 2 g of NaCl, 7 mL of concentrated hydrochloric acid followed by adding ddH_2_O to achieve a final volume of 1 L. Briefly, 0.3 g of PMS (K_2_SO_4_·MgSO_4_·6H_2_O), 0.117 g of potassium sulfate (K_2_SO_4_), and 0.162 g of magnesium sulfate (MgSO_4_·7H_2_O) were added separately to three individual samples of pig gastric juice (100 mL, 37 °C, K^+^ 0.05238 g/mL, Mg^2+^ 0.0162 g/mL). Then, the dissociation solutions of the three groups at 10 s, 20 s, 30 s, 1 min, 5 min, and 10 min were collected to measure the concentrations of K^+^ and Mg^2+^ in the dissociation solutions, thereby allowing for a comparison of the dissociation rates of PMS with those of magnesium sulfate and potassium sulfate at various time points. The concentrations of K^+^ and Mg^2+^ were measured using the potassium (K) detection kit (Cat No. C001-2-1) and magnesium (Mg) detection kits (Cat No. C005-1-1) were provided by Nanjing Jiancheng Biotechnology Co., Ltd. (Nanjing, China).

#### 2.1.2. In Vitro Ussing Chamber Assay

For the preparation of intestinal mucosal tissue used in Ussing chamber assays, 21-day-old weaned piglets were selected, fasted, and anesthetized to harvest segments of the duodenum, jejunum, and ileum, with 3 replicates for each segment. The 3 replicates of intestinal segments were employed for Ussing chamber assays involving PMS, potassium sulfate, and magnesium sulfate, following the protocols described previously [37]. Each intestinal segment was cut to approximately 2 cm^2^, washed with physiological saline, and subsequently placed in ice-cold Krebs solution for 5 min, followed by removal of the serosal layers. After pre-incubating the intestinal mucosa for 15 min in Krebs solution, 5 mL each of PMS-Krebs, potassium sulfate-Krebs, and magnesium sulfate-Krebs solutions were added to the mucosal side (diffusion chamber), and 5 mL of Krebs solution was added to the serosal side (absorption chamber). After 60 min of the experiment, 2 mL of solution samples were taken from the diffusion and absorption chambers to determine K^+^ and Mg^2+^ absorption in the intestinal tract of weaned piglets. The Krebs buffer stock solution (1 L) was composed of 7.35 g CaCl_2_·2H_2_O, 7.01 g KCl, 4.88 g MgCl_2_·6H_2_O, and 3.74 g NaH_2_PO_4_·2H_2_O. Krebs buffer (2 L) was composed of 100 mL of Krebs buffer stock solution and 13.76 g NaCl, 4.20 g NaHCO_3_, and 3.96 g glucose. The initial K^+^ and Mg^2+^ ion concentrations were determined as K^+^ of 7.53 mmol/L and Mg^2+^ of 1.27 mmol/L in PMS-Krebs, K^+^ of 7.54 mmol/L, Mg^2+^ of 1.25 mmol/L in potassium sulfate-Krebs, and K^+^ of 4.03 mmol/L, Mg^2+^ of 0.47 mmol/L in magnesium sulfate-Krebs. K^+^ and Mg^2+^ ion concentrations were determined according to the manufacturer’s instructions, as mentioned above. The contents of K^+^ or Mg^2+^ ion (mmol/L) are calculated using the formula as previously described [37]: (initial ion concentration in diffusion chamber + initial ion concentration in absorption chamber) − (termination ion concentration in diffusion chamber + termination ion concentration in absorption chamber).

### 2.2. Experiment 2

#### 2.2.1. Animals, Diets, and Experimental Design

A total of 216 healthy 21-day-old weaned piglets (Duroc × Landrace × Large White, initial body weight of 7.52 ± 0.02 kg) were randomly allotted into 6 groups. Each group contained 6 replicates of 6 piglets (3 castrated males and 3 females) per replicate. Piglets were fed a basal diet supplemented with 0, 0.15, 0.30, 0.45, 0.60, and 0.75% PMS. The experimental design was consistent with that previously described [18]. PMS products (K_2_SO_4_·MgSO_4_·6H_2_O, containing 21% K and 6.5% Mg), sourced from Qinghai Salt Lake, were produced and provided by Qinghai Lanhushancheng Bio-tech Co., Ltd. (Xining, China). The compositions of the basal diets (Table 1) for phase 1 (1–28 d) and phase 2 (29–42 d) were formulated to meet or exceed the nutritional requirements for 7–11 kg and 11–25 kg weaned piglets, respectively, as recommended by NRC (2012) [9]. All pigs were housed in an environmentally controlled room featuring slatted plastic flooring and an effective automatic ventilation system that maintained a temperature of 25–28 °C and relative humidity of 55–65%. The pigs had free access to food and water throughout the trial.

#### 2.2.2. Sample Collection

On the last three days (days 39–41) before the end of the experiment (day 42), fecal samples of approximately 20–30 g per piglet were collected from each replicate (*n* = 6) continuously over a period of three days after feeding. The collected fecal samples within each replicate from the three days were mixed and stored in sealed bags at −20 °C until analysis of K^+^ and Mg^2+^ contents in the feces.

On the morning of day 42, all piglets were weighed after fasting for 12 h, and one piglet with a similar body weight in each replicate was selected for blood sample collection from the anterior vena cava. Blood samples (10 mL) were maintained at room temperature for 30 min, followed by centrifugation at 3000 r/min at 4 °C for 10 min to obtain serum samples for the determination of immune parameters and K^+^ and Mg^2+^ contents. After blood collection, the piglets were euthanized for tissue collection. The jejunum and colon mucosa samples were gently scraped using a glass slide, rapidly frozen in liquid nitrogen, and stored at −80 °C for subsequent analysis of intestinal gene expression.

#### 2.2.3. Determinations of Serum Immune Parameters

The serum levels of immunoglobulin A (IgA), immunoglobulin M (IgM), immunoglobulin G (IgG), interleukin (IL)-1β, IL-2, IL-8, IL-10, and tumor necrosis factor-α (TNF-α) were determined according to the manufacturer’s instructions (Enzyme-linked Biotechnology Co., Ltd., Shanghai, China) and measured using a multi-functional microplate reader (SynergyTM H1, BioTek Instruments, Inc., Winooski, VT, USA).

#### 2.2.4. RNA Extraction, cDNA Synthesis, and qPCR Assay

Total RNA was extracted from the jejunal and colonic mucosa of piglets using the TRIzol method. The purity and concentration of the extracted total RNA samples were assessed using a NanoDrop-ND1000 spectrophotometer (Thermo Fisher Scientific Inc., Dreieich, Germany) with an A260/280 of 1.8–2.0. cDNA synthesis from total RNA (1 μg) was performed using the PrimeScript^®^ RT reagent kit with a gDNA eraser (Takara Biotechnology Co., Dalian, China). The cDNA template was diluted with DEPC water and stored separately at −20 °C until use. The fluorescence quantitative PCR reaction procedure included the pre-denaturation at 95 °C for 30 s, denaturation at 95 °C for 15 s, annealing at 60 °C for 30 s, and extension at 72 °C for 30 s, followed by 40 cycles, concluding with a final denaturation at 95 °C. Forward and reverse primer sequences (Table 2) were designed using Primer Premier 5.0 software (Applied Biosystems, Waltham, MA, USA) and synthesized by the Guangzhou Brand of Shanghai Sangon Biotech Co. (Guangzhou, China). The relative mRNA expression of the target genes was normalized to that of the housekeeping gene (*β-actin*) and calculated using the 2^−∆∆ct^ method.

#### 2.2.5. Statistical Analysis

For the data from the in vitro study, the Student’s *t*-test was used to compare the intestinal dissociation and absorption of K^+^ from PMS and K_2_SO_4_ or Mg^2+^ from PMS and MgSO_4_. Moreover, data analysis for the in vivo study was performed using one-way ANOVA with SPSS software (IBM SPSS Statistics 22) after checking the homogeneity of variance. Regression models of linear and quadratic polynomial contrasts were used to determine the dose effects. Duncan’s method was used for multiple comparisons. Results are presented as mean ± standard error (SE). A *p*-value less than 0.05 indicates a statistically significant difference, while a *p*-value between 0.05 and 0.10 suggests a potential trend worth further investigation. Spearman correlation analysis was used to analyze the relationship between AQP expression and diarrhea rate at 29–42 days in piglets. Figures were created using GraphPad Prism software (Version 8.0.1).

## 3. Results

### 3.1. The Dissociation and Absorption of K^+^ and Mg^2+^ in PMS, Potassium Sulfate, and Magnesium Sulfate

There was no significant difference in the dissociation rates of PMS, potassium sulfate, and magnesium sulfate in the simulated pig gastric juice (*p* > 0.05) (Figure 1). Similarly, there were no significant differences in the absorption rates of PMS, potassium sulfate, and magnesium sulfate in the duodenum, jejunum, and ileum of weaned piglets (*p* > 0.05) (Figure 2). However, the Ussing chamber assay revealed that the absorption of potassium and magnesium ions in the jejunum and ileum was significantly greater than that in the duodenum of the weaned piglets (*p* < 0.05) (Figure 3).

### 3.2. Effects of Dietary Supplementation with PMS on Serum and Fecal Potassium and Magnesium Ion Concentrations in Weaned Piglets

As shown in Figure 4, the addition of PMS linearly increased serum potassium ion concentration (*p* < 0.05) but had no significant effect on serum magnesium ion concentration in weaned piglets (*p* > 0.05). Furthermore, the addition of 0.30%, 0.45%, 0.60%, and 0.75% PMS significantly increased magnesium ion concentrations in the feces of weaned piglets (*p* < 0.05).

### 3.3. The Effect of Dietary Supplementation with PMS on Serum Immune Indicators of Weaned Piglets

As shown in Table 3, the addition of PMS resulted in a linear increase in serum concentrations of IgG and IL-2 while simultaneously leading to a linear decrease in serum IL-1β levels in weaned piglets (*p* < 0.05). Moreover, the addition of 0.60% or 0.75% PMS significantly increased the serum IL-10 levels in weaned piglets (*p* < 0.05).

### 3.4. Effect of Dietary Supplementation with PMS on Intestinal Expression of NLRP3 Inflammasome and Cytokines in Weaned Piglets

As shown in Figure 5, dietary supplementation with PMS significantly reduced the mRNA expression of *NLRP3* in the jejunum of piglets compared with that in the control group (*p* < 0.05). Additionally, the mRNA expression of *ASC* in the jejunum exhibited a quadratic and significant decrease (*p* < 0.05), while the mRNA expression of *Caspase-1* and *IL-1β* in the jejunum demonstrated a significant linear decrease due to dietary PMS supplementation (*p* < 0.05). Furthermore, the addition of PMS did not significantly affect the mRNA expression of *NLRP3* in the colon (*p* > 0.05), but it linearly reduced the mRNA expression of *ASC* and *Caspase-1* in the colon (*p* < 0.05) and significantly decreased the mRNA expression of *IL-1β* in the colon of weaned piglets (*p* < 0.05).

### 3.5. Effect of Dietary Supplementation with PMS on Intestinal Expression Levels of Potassium and Magnesium Ion Channels in Weaned Piglets

As shown in Figure 6, varying levels of dietary PMS supplementation linearly reduced the mRNA expression of *KCNK5* and *KCNK6* in the jejunum of weaned piglets (*p* < 0.05). However, there was no significant effect on the mRNA expression of *KCNK12* in the jejunum of weaned piglets (*p* > 0.05). Additionally, dietary supplementation with PMS linearly decreased the mRNA expression of *KCNK5* in the colon (*p* < 0.05), while no significant effects were observed on the mRNA expression of *KCNK6* and *KCNK12* in the colon of weaned piglets (*p* > 0.05).

As shown in Figure 7, different doses of dietary PMS significantly reduced the mRNA expression of *TRPM6* in the jejunum (*p* < 0.05) and linearly decreased the mRNA expression of *TRPM7* in the jejunum of weaned piglets (*p* < 0.05). In contrast, there was no significant change in the expression of *MagT1* mRNA in the jejunum of weaned piglets (*p* > 0.05). Furthermore, dietary supplementation with 0.60% and 0.70% PMS significantly increased the mRNA expression of *TRPM6* in the colon (*p* < 0.05) and linearly enhanced the mRNA expression of TRPM7 in the colon (*p* < 0.05), while there was no significant change in the mRNA expression of *MagT1* in the colon of weaned piglets (*p* > 0.05).

### 3.6. Effect of Dietary Supplementation with PMS on Intestinal Expression of AQPs in Weaned Piglets

As shown in Figure 8, the addition of PMS resulted in a linear increase in the mRNA expression of *AQP8* in the jejunum of weaned piglets (*p* < 0.05). There was no significant effect on the mRNA expression of *AQP1*, *AQP3,* and *AQP7* in the jejunum (*p* > 0.05). Furthermore, PMS did not significantly affect the mRNA expression of *AQP1* and *AQP7* in the colon of weaned piglets (*p* > 0.05). However, the addition of 0.45% and 0.60% PMS significantly increased the mRNA expression of *AQP3* in the colon (*p* < 0.05), and 0.75% PMS significantly increased the mRNA expression of *AQP8* in the colon of weaned piglets (*p* < 0.05). Moreover, the mRNA expression of *AQP3* and *AQP8* in the colon of weaned piglets (Figure 9) was significantly negatively correlated with the incidence of diarrhea observed at 29–42 days of the experiment (*p* < 0.05).

## 4. Discussion

Generally, the digestive function and immune system of weaned piglets are not yet fully developed, and the small intestinal barrier and absorptive functions deteriorate shortly after weaning [38]. Consequently, weaned piglets are prone to stunted growth and develop diarrhea characterized by disturbances in intestinal fluid and ion balance [2]. Notably, the requirement for minerals may need to be adjusted to reflect the increased production levels of swine in modern intensive production systems [4]. Our previous research demonstrated that dietary PMS supplementation significantly improves growth performance, enhances antioxidant capacity, and reduces the incidence of post-weaning diarrhea in piglets [18]. However, the dissociation and absorption of PMS in the gastrointestinal tract of weaned piglets and the potential effect of dietary PMS supplementation on the intestinal expression of genes involved in water and ion transport remain unclear.

The Ussing chamber is a representative model for investigating the mechanisms underlying the absorption and transport of nutrients and drugs without interference from intestinal digesta [39]. Potassium is crucial for maintaining electrolyte balance and acid-base equilibrium within the body, and the interrelationships between magnesium and potassium homeostasis are vital for the overall health of the organism [40]. The absorption of potassium is typically influenced by dietary potassium intake and the intestinal transport processes of potassium [41]. Consequently, the ability of the gut to sense potassium intake plays an important role in regulating potassium homeostasis, which encompasses both the intestinal absorption and secretion of potassium [42,43,44]. Hegazy et al. demonstrated that active net absorption occurs in both the jejunum and ileum of piglets, as evidenced by measurements of water, sodium, and potassium absorption, revealing a significant correlation between water and electrolyte transport [45]. Similarly, the present study indicated that the jejunum and ileum, rather than the duodenum, are the primary sites for the absorption of potassium and magnesium ions in the intestines of weaned piglets, as determined using the Ussing chamber assay. However, this study found that there were no significant differences in the potassium and magnesium ion concentrations across various intestinal mucosa and serous sides of weaned piglets when comparing PMS with K_2_SO_4_ or MgSO_4_, nor in the dissociation rates of PMS, potassium sulfate, and magnesium sulfate, as determined with pig gastric juice. An earlier investigation using a Ussing chamber assay revealed that increased jejunal K^+^ secretion and impaired apical K^+^ absorption were observed in piglet rotavirus enteritis, potentially leading to elevated K^+^ loss in the feces [46]. Moreover, fecal K^+^ excretion has been shown to be significantly increased during episodes of diarrhea [41]. Although we were unable to detect an increase in fecal K^+^ concentration in weaned piglets-fed diets supplemented with PMS, both serum K^+^ and fecal Mg^2+^ concentrations exhibited a linear increase in response to dietary PMS supplementation. This finding is consistent with previous reports indicating that Mg absorption might be influenced by dietary K concentrations in dairy cows [47].

Aquaporins (AQPs) are a family of small membrane proteins comprising at least 13 subtypes (AQP0-AQP12), which are mainly expressed in the plasma membranes of numerous cell types [29]. These proteins are crucial for the maintenance of fluid balance and electrolyte homeostasis [29]. Alterations in the expression and localization of AQPs within the gastrointestinal tract have been shown to correlate with the pathology of gastrointestinal disorders [48], with AQP3 being closely involved in the regulation of intestinal health [49]. In the present study, we observed that the mRNA expression of *AQP3* and *AQP8* in the colonic mucosa was significantly upregulated by dietary PMS supplementation. This finding aligns with a previous study indicating that magnesium sulfate enhances intestinal AQP3 expression in HT-29 cells [50]. Similarly, another in vitro study revealed that magnesium ions upregulated both AQP3 mRNA and protein expression in Caco-2 cells [51]. A previous study confirmed the critical role of AQPs in IL-1β-induced inflammation, which suggests an important relationship between AQP expression and cytokines [52]. Moreover, several AQP subtypes, including AQP1, AQP3, and AQP4, have been identified as important markers of inflammation [53]. Indeed, AQP3 could promote M2 macrophage polarization via the peroxisome proliferator-activated receptor-γ (PPAR-γ)/ nuclear factor kappa B (NF-κB) axis to modulate the production of pro-inflammatory cytokines [54]. Additionally, TNF-α decreases AQP3 expression in intestinal epithelial cells by inhibiting constitutive transcription of the AQP3 promoter [55]. Consistent with this, we previously reported that jejunal TNF-α levels were linearly reduced by dietary PMS supplementation [18]. Furthermore, the current study demonstrated a strong negative correlation between intestinal AQP3 and AQP8 expression and the incidence of diarrhea in weaned piglets. This observation may, at least in part, explain the reduced incidence of diarrhea in weaned piglets following dietary PMS supplementation [18], probably through an AQP-dependent mechanism.

Potassium and magnesium are essential minerals that play critical roles in maintaining immune homeostasis. Increasing evidence has demonstrated the efficacy of potassium supplementation in enhancing broiler performance through its incorporation into drinking water or feed [15,16,56]. Furthermore, the benefits of magnesium supplementation in livestock and poultry diets have been acknowledged in previous studies [13,14,47,57,58,59,60]. This is attributable to the fact that magnesium is a vital mineral necessary for numerous physiological processes, serving as a cofactor in over 300 enzymatic reactions, including ionic balance, mitochondrial function, ATP production, energy metabolism, and synthesis of proteins and nucleic acids [6,61]. Additionally, magnesium plays an important role in immunoglobulin synthesis and the immune response [62]. Notably, patients with inflammatory bowel disease have exhibited a significant magnesium deficit, with serum magnesium levels inversely correlated with disease activity [63]. A deficiency of magnesium in the body can disrupt the inflammatory response, increase the risk of infections, and impair cell-mediated immunity and IgG synthesis, potentially leading to temporary or long-term immune dysfunction [64,65]. Thus, maintaining adequate levels of magnesium in the diet is essential for optimal immune function and the regulation of inflammation [64,66] and contributes to the improvement of the lifespan of animals [61]. Dietary magnesium sulfate supplementation has been shown to linearly enhance the IgG level in plasma on the day of farrowing, quadratically increase IgA levels in colostrum, and linearly increase IgA levels in the milk of sows [67]. However, there are relatively few reports on the effects of dietary PMS supplementation on immune function and intestinal health in weaned piglets. Our previous study also demonstrated that the levels of IgM were increased in the intestines of weaned piglets through dietary PMS supplementation at 0.15–0.75% [18]. Consistently, the dietary PMS supplementation resulted in a significant increase in serum IL-10 levels in weaned piglets in the current study, indicating that dietary PMS supplementation is beneficial for immune regulation in weaned piglets.

The NLRP3 inflammasome complex comprises NLRP3, adapter protein ASC, and inflammatory protease caspase-1. As a crucial component of the innate immune system, NLRP3 inflammasome plays a significant role in the regulation of inflammation and immune responses [25]. The NLRP3 inflammasome is activated under conditions of stress and inflammation [19]. A recent study showed that weaning stress significantly decreases the mRNA expression of *NLRP3* in the jejunum of low-birth-weight piglets [68]. In this study, we found for the first time that dietary PMS supplementation significantly decreased the intestinal expression of the NLRP3 inflammasome (*NLRP3*, *ASC*, and *Caspase-1*) and the expression of its downstream cytokine IL-1β in weaned piglets. This was in accordance with a previous study that showed that magnesium sulfate regulates lipopolysaccharide (LPS)-induced fetal membrane inflammation [69]. Similarly, López-Baltanás et al. [70] demonstrated that magnesium supplementation could mitigate inflammation in rats, as evidenced by a reduction in pro-inflammatory cytokines (TNF-α, IL-6, and IL-1β) associated with chronic kidney disease. Another study conducted in both cultured macrophages and mice demonstrated that magnesium-enriched deep-sea water inhibited NLRP3 inflammasome activation and mitigated inflammation [71]. Moreover, previous research has demonstrated that magnesium isoglycyrrhizinate inhibits LPS-induced inflammation and oxidative stress by suppressing the levels of TNF-α, IL-6, IL-1β, IL-8, NO and inducible nitric oxide synthase (iNOS) through the NF-κB and mitogen-activated protein kinase (MAPK) pathways in LPS-treated RAW264.7 cells [72]. Additionally, a high-potassium environment mitigated the increase in IL-1β in bone marrow-derived macrophages (BMDMs) induced by ATP and LPS [73,74,75]. Importantly, potassium ion efflux has been found to be associated with oxidative stress and inflammation, functioning as a common trigger for NLRP3 inflammasome activation through the augmentation of potassium channels [73,74,76]. This was supported by another study demonstrating that activation of the NLRP3 inflammasome in macrophages was lower in two-pore potassium (K2P) channel TREK-1 knock-out mice than in wild-type mice [77]. Thus, the observed alterations in immune parameters indicated that PMS may assist weaned piglets in better managing weaning stress and potential pathogen challenges following weaning. By modulating the expression of the NLRP3 inflammasome, dietary PMS may contribute to the reduction of intestinal inflammation and maintenance of gut health, thereby enhancing the overall well-being and growth performance of weaned piglets.

Potassium channels are transmembrane proteins that selectively facilitate the flow of potassium ions along an electrochemical gradient [78]. Previous studies have demonstrated that KCNK5 deficiency significantly reduces cell apoptosis [79], while the absence of KCNK6 inhibits macrophage proliferation, and KCNK6 knock-out mice exhibit reduced concentrations of IL-1β, IL-18, TNF-α, and myeloperoxidase (MPO) after LPS treatment [73]. In this study, we observed that dietary supplementation with higher doses of PMS resulted in a decrease in intestinal *KCNK5* in the jejunum and colon, as well as *KCNK6* in the jejunum. This finding is supported by a meta-analysis of case-control studies indicating that a high intake of dietary calcium, magnesium, and potassium is negatively correlated with the incidence of colorectal cancer [80]. Interestingly, the mRNA expression of voltage-gated potassium (Kv) channels (Kv1.1 and Kv1.5) in the jejunal and ileal mucosa of weaned piglets was significantly downregulated on days 5 and 7 post-weaning [81]. Furthermore, a magnesium-deficient diet led to hypomagnesemia and aggravated dextran sodium sulfate (DSS)-induced colitis, which was accompanied by compromised intestinal magnesium absorption and reduced TRPM6 expression [63]. Severe hypomagnesemia is associated with reduced magnesium absorption by downregulating TRPM6-mediated magnesium influx into Caco-2 cells [82]. However, dietary magnesium supplementation can ameliorate disease symptoms and restore mucosal function in a murine model of DSS-induced colitis [63,83]. This restoration effect may be partially attributed to the upregulation of TRPM 6 expression by dietary magnesium supplementation [63]. Indeed, TRPM6 and TRPM7 are directly involved in epithelial magnesium reabsorption for Mg^2+^ uptake in vertebrate epithelial cells, thereby maintaining magnesium homeostasis in the body, which leads to increased cell proliferation and migration [84,85,86]. Consistently, the present study found that dietary PMS supplementation increased colonic *TRPM6* expression but decreased jejunal *TRPM6* expression in weaned piglets. The discrepancy in PMS on this gene expression may be associated with the varied expression abundance and function of TRPM6 in different intestinal segments, which requires further investigation. Collectively, these findings indicate that dietary PMS supplementation at different dosages significantly influenced the expression of *AQP3* and *AQP8*, as well as potassium and magnesium ion channels in the intestines of weaned piglets. However, the limitations of the present study are that only the mRNA expression of these water and ion channels was determined after dietary PMS addition, and investigations into their protein expression, as well as the potential molecular mechanism by which PMS affects the expression of these genes are lacking. Consequently, based on the cost-effectiveness considerations of PMS and the findings from our current and previous studies [18], it is recommended to supplement conventional corn-soybean diets with 0.30% PMS to optimize the health of weaned piglets. This finding suggests that dietary PMS supplementation may enhance the efficiency of water and electrolyte transport in the gastrointestinal tract, which is crucial for maintaining hydration and preventing diarrhea in weaned piglets.

## 5. Conclusions

In conclusion, the present study demonstrated that dietary PMS supplementation could improve immune function and modulate the expression of the NLRP3 inflammasome, as well as AQPs, K, and Mg ion channels, which might be important for controlling the post-weaning diarrhea in piglets. The recommended supplemental dosage of PMS in the corn-soybean basal diet for weaned piglets is 0.30%. These findings suggest that PMS could serve as a valuable dietary supplement to enhance the health and well-being of piglets, particularly during the critical post-weaning period. These results provide significant insights into the potential mechanisms underlying the beneficial effects of PMS supplementation on the health and physiological functions of weaned piglets. However, further research is necessary to explore the potential role and mechanisms of PMS on immune function and intestinal health in pigs across different physiological stages or stress conditions as well as to determine the optimal dosage and duration of PMS supplementation for maximum benefits.

## Figures and Tables

**Figure 1 animals-15-01751-f001:**
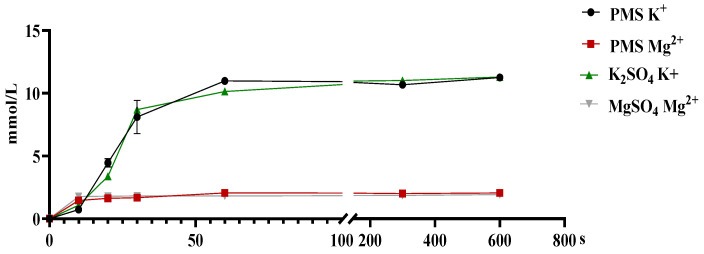
Dissociation concentrations of PMS, K_2_SO_4_, and MgSO_4_ in simulated pig gastric juice. Solutions of PMS, K_2_SO_4_, and MgSO_4_ were added separately to three individual samples of pig gastric juice (100 mL, 37 °C, K^+^ 0.05238 g/mL, Mg^2+^ 0.0162 g/mL). Then, the dissociation solutions of the three groups (*n* = 3) were collected to measure the concentrations of K^+^ and Mg^2+^ in the dissociation solutions at various time points (10 s, 20 s, 30 s, 1 min, 5 min, and 10 min). PMS, potassium magnesium sulfate.

**Figure 2 animals-15-01751-f002:**
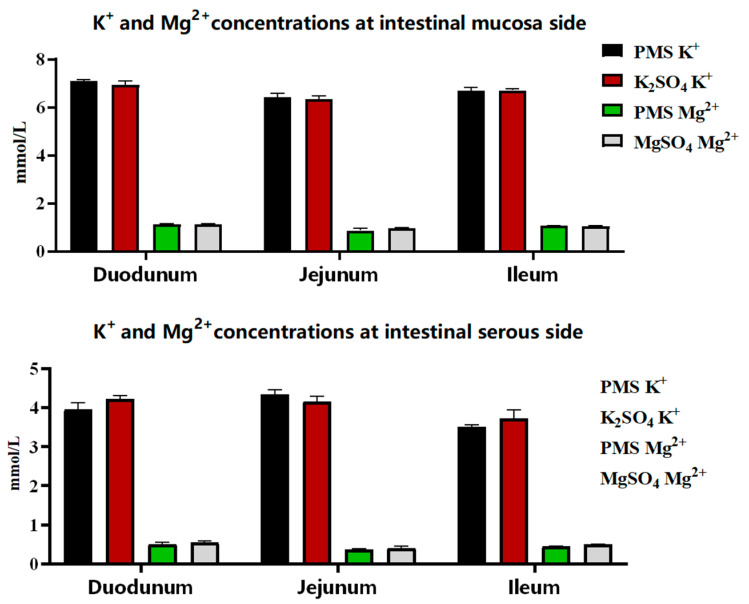
Residual concentrations of potassium and magnesium ions in different intestinal mucosa and serous sides of weaned piglets. The duodenum, jejunum, and ileum segments (2 cm^2^) from 21-day-old weaned piglets were used to investigate the absorption of PMS, K_2_SO_4_, and MgSO_4_ in Krebs solution using Ussing chamber assay (*n* = 3). PMS, potassium magnesium sulfate.

**Figure 3 animals-15-01751-f003:**
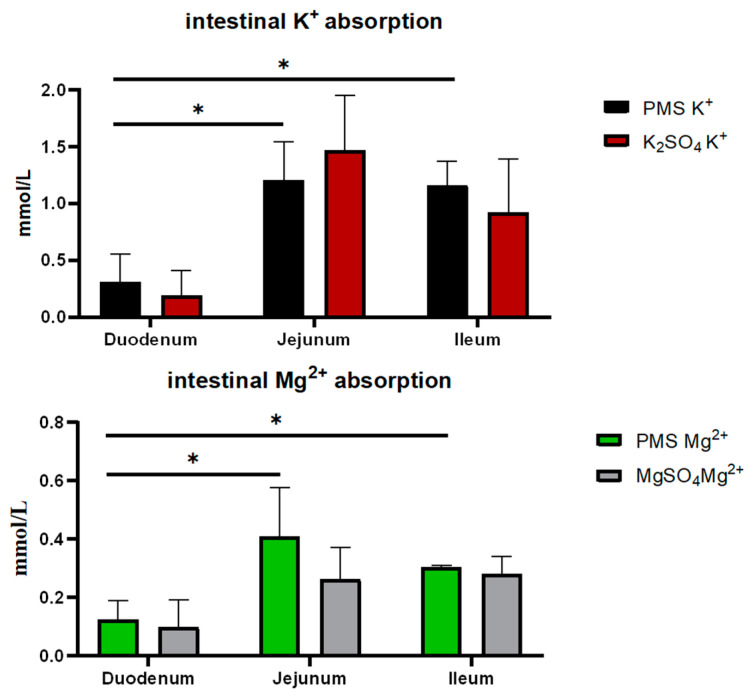
Potassium and magnesium absorption by PMS, K_2_SO_4_, and MgSO_4_ in the intestine. PMS: potassium magnesium sulfate. * indicates significant difference between the two groups (*p* < 0.05) (*n* = 3).

**Figure 4 animals-15-01751-f004:**
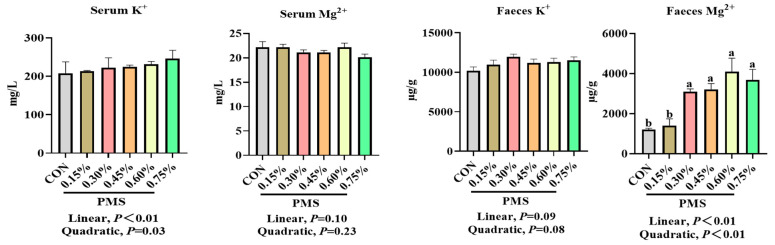
Effect of dietary PMS supplementation on the contents of potassium and magnesium ions in the serum and feces of weaned piglets. CON, basal diet; 0.15%, diet containing 0.15% PMS; 0.3%, diet containing 0.3% PMS; 0.45%, diet containing 0.45% PMS; 0.6%, diet containing 0.6% PMS; 0.75%, diet containing 0.75% PMS. PMS: potassium magnesium sulfate. ^a,b^ Means in columns without a common superscript letter differ significantly at *p* < 0.05. Data are presented as mean ± SE (*n* = 6).

**Figure 5 animals-15-01751-f005:**
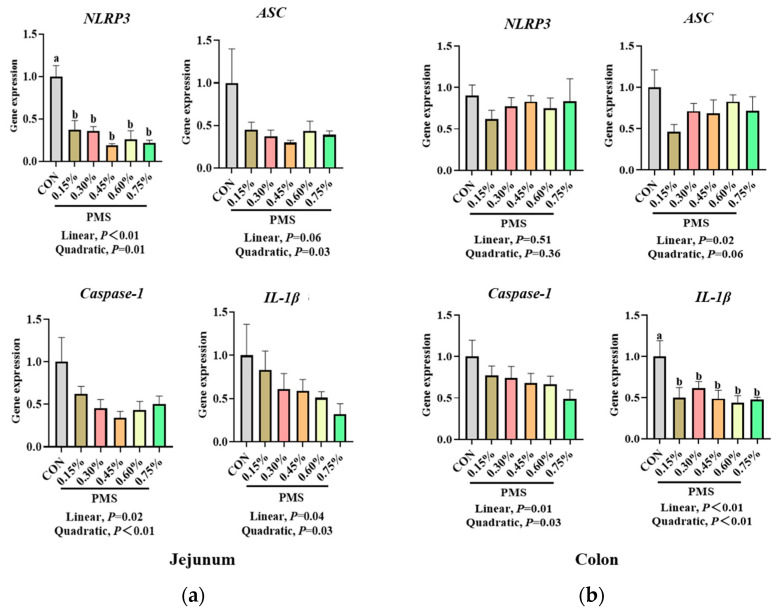
Effect of dietary PMS supplementation on intestinal expression of NLRP3 inflammasome in weaned piglets. (**a**) Jejunum. (**b**) Colon. PMS, Potassium–magnesium sulfate; *NLRP3*, NOD-like receptor thermal domain-associated protein 3; *ASC*, apoptosis-associated speck-like protein containing a CARD; *Caspase-1*, cysteine protease 1; *IL-1β*, interleukin-1β. CON, basal diet; 0.15%, diet containing 0.15% PMS; 0.3%, diet containing 0.3% PMS; 0.45%, diet containing 0.45% PMS; 0.6%, diet containing 0.6% PMS; 0.75%, diet containing 0.75% PMS. ^a,b^ Means in columns without a common superscript letter differ significantly at *p* < 0.05. Data are presented as mean ± SE (*n* = 6).

**Figure 6 animals-15-01751-f006:**
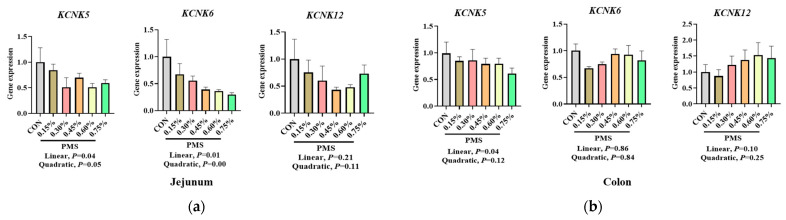
Effect of PMS on mRNA expression of potassium channels in the jejunum and colon of weaned piglets. (**a**) Jejunum. (**b**) Colon. PMS, potassium magnesium sulfate; *KCNK5*, potassium channel subfamily K member 5; *KCNK6*, potassium channel subfamily K member 6; *KCNK12*, potassium channel subfamily K member 12. CON, basal diet; 0.15%, diet containing 0.15% PMS; 0.3%, diet containing 0.3% PMS; 0.45%, diet containing 0.45% PMS; 0.6, diet containing 0.6% PMS; 0.75%, diet containing 0.75% PMS. Data are presented as mean ± SE (*n* = 6).

**Figure 7 animals-15-01751-f007:**
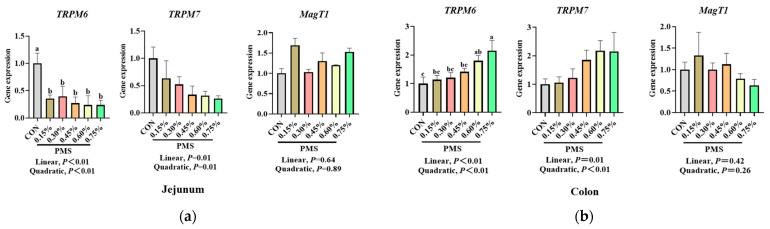
Effect of PMS on mRNA expression of magnesium channels in the jejunum and colon of weaned piglets. (**a**) Jejunum. (**b**) Colon. PMS, Potassium–magnesium sulfate; *TRPM6*, transient receptor potential cation channel, subfamily M, member 6; *TRPM7*, transient receptor potential cation channel, subfamily M, member 7; *MagT1*, magnesium transporter 1. CON, basal diet; 0.15%:, diet containing 0.15% PMS; 0.3%, diet containing 0.3% PMS; 0.45%, diet containing 0.45% PMS; 0.6%, diet containing 0.6% PMS; 0.75%, diet containing 0.75% PMS. ^a,b,c^ Means in columns without a common superscript letter differ at *p* < 0.05. Data are presented as mean ± SE (n = 6).

**Figure 8 animals-15-01751-f008:**
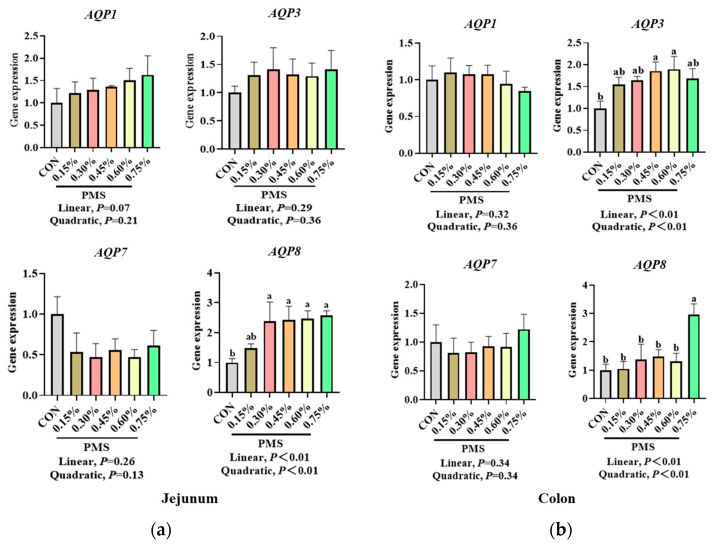
Effect of PMS on mRNA expression of aquaporins in the jejunum and colon of weaned piglets. (**a**) Jejunum. (**b**) Colon. PMS, Potassium–magnesium sulfate; *AQP1*, aquaporin 1; *AQP3*, aquaporin 3; *AQP7*, aquaporin 7; *AQP8*, aquaporin 8. CON, basal diet; 0.15%, diet containing 0.15% PMS; 0.3%, diet containing 0.3% PMS; 0.45%, diet containing 0.45% PMS; 0.6%, diet containing 0.6% PMS; 0.75%, diet containing 0.75% PMS. ^a,b^ Means in columns without a common superscript letter differ at *p* < 0.05. Data are presented as mean ± SE (*n* = 6).

**Figure 9 animals-15-01751-f009:**
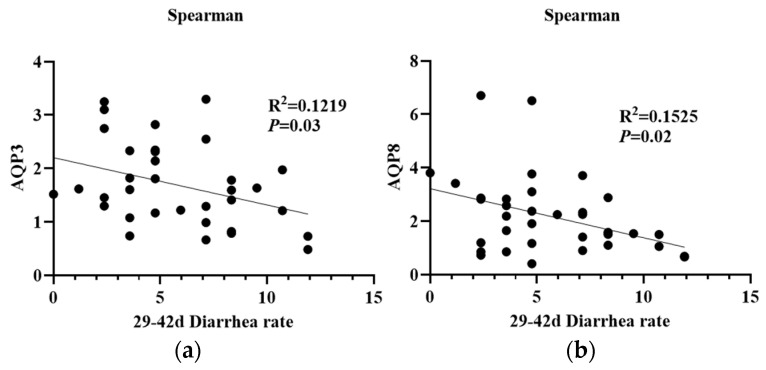
Spearman’s correlation analysis between the incidence of diarrhea and intestinal aquaporin expression in weaned piglets. (**a**) Correlation between intestinal aquaporin-3 (AQP3) and diarrhea rate in piglets at 29–42 d. (**b**) Correlation between intestinal aquaporin-8 (AQP8) and diarrhea rate in piglets at 29–42 d. PMS, potassium magnesium sulfate.

**Table 1 animals-15-01751-t001:** Composition and nutrition level of the basal diet (air-dried).

Item	1–28 Days	29–42 Days
Ingredient, %		
Corn	34.05	56.18
Expanded corn	12.00	13.00
Fermented soybean meal	10.00	10.00
Soybean meal	5.00	6.00
Expanded soybean	11.00	3.67
Fish meal	3.00	3.00
Whey powder	15.00	~
Whey protein concentrate	1.00	~
Soybean oil	1.50	1.00
Sucrose	2.00	2.00
Calcium citrate	1.40	~
CaCO_3_	~	1.10
CaHPO_4_	0.60	0.60
L-lys-HCl	0.60	0.60
DL-Met	0.15	0.15
L-Thr	0.20	0.20
L-Trp	0.05	0.05
NaCl	0.30	0.30
50% Choline chloride	0.15	0.15
Premix ^1^	2.00	2.00
Total	100	100
Nutrient level ^2^		
DE, MJ/kg	15.06	14.60
CP, %	18.70	17.20
SID Lys, %	1.54	1.35
SID Met, %	0.47	0.45
SID Met+Cys, %	0.78	0.74
SID Thr, %	0.93	0.83
SID Trp, %	0.27	0.24
Ca, %	0.70	0.76
Total P, %	0.55	0.53
STTD P, %	0.36	0.31
K, %	1.03	0.68
Mg, %	0.19	0.15
Na, %	0.30	0.16
Cl, %	0.56	0.36
dEB, mmol/kg	211	125

^1^ The premix provided the following per kilogram of diets: VA 11,000 IU, VD 31,100 IU, VE 30 IU, VK 2.5 mg, VB_12_ 0.1 mg, VB_1_ 3 mg, VB_2_ 10 mg, Niacin 150 mg, D-pantothenic acid 60 mg, Folic acid 1.5 mg, VB_6_ 8 mg, Biotin 0.4 mg, Fe 120 mg, Cu 6 mg, Mn 4 mg, Zn 100 mg, I 0.14 mg, Se 0.3 mg. ^2^ Nutrient levels were calculated values, except that the CP, Ca, total P, available P, K, and Mg were determined values.

**Table 2 animals-15-01751-t002:** Sequences and accession numbers of primers for the targeted genes.

Primer ^1^	Sequence (5′ to 3′)	NCBI Access Number
*β-actin*	F: TCTGGCACCACACCTTCT R: TGATCTGGGTCATCTTCTCAC	XM_021086047.1
*NLRP3*	F: CCTTCAGGCTGATTCAGGAG R: GACTCTTGCCGCTATCCATC	NM_001256770.2
*Caspase-1*	F: ATCTCACCGCTTCGGACATGGCTAT R: GTATTTCTTCCCACAAATGCCAGCC	NM_214162.1
*ASC*	F: GCCGACGAGCTCAAGAAGTT R: TCCTTCATGCCGATGTCACG	AB873106.1
*IL-1β*	F: TCTGCATGAGCTTTGTGCAAG R: ACAGGGCAGACTCGAATTCAAC	XM_021085847.1
*KCNK5*	F: GGCACGTATCTCACCATCCC R: TGATGGCCTCTTCCCTACGA	XM_001928254.4
*KCNK6*	F: GGGAGATGCAGAGGCTTGTT R: CCAGTTTCAATCACCCCCGA	XM_003127105.4
*KCNK12*	F: AACGTAGCACCAACTAGCGG R: CACTTCCTCGCTTCCCTGTTT	XM_005674606.3
*TRPM6*	F: TGGTGGAGCATATCGCAGTAG R: AGGCTGGGCAGTTCTAATGA	XM_021064975.1
*TRPM7*	F: CTCTCACGTGGTCTGTCTTCA R: ATTGTGCGACCAACTCCTCC	XM_021095614.1
*MagT1*	F: TATCGTATCCAAAGGTGGGGG R: GTAGGGAGACAGAATACACCTGA	XM_003135205.6
*AQP1*	F: AGACACTCTGACAAGCTGGC R: GTCAAGGGAGTGGGTGATGG	XM_021078524.1
*AQP3*	F: TGACCTTCGCTATGTGCTTCC R: GTCCAAGTGTCCAGAGGGGTAG	NM_001110172.1
*AQP7*	F: CGTGACCTCTACCCACAACC R: TGGGGAGGGGGTCACAAATA	XM_013980184.2
*AQP8*	F: AGGAGGGCTCATCAGGTTCT R: CTCAGCTTCACCGTCCCTTT	NM_001112683.1

^1^ *NLRP3*, NOD-like receptor thermal domain-associated protein 3; *ASC*, apoptosis-associated speck-like protein containing a CARD; *Caspase-1*, cysteine protease 1; *IL-1β*, interleukin-1β; KCNK5, potassium channel subfamily K member 5; *KCNK6*, potassium channel subfamily K member 6; *KCNK12*, potassium channel subfamily K member 12; *TRPM6*, transient receptor potential cation channel, subfamily M, member 6; *TRPM7*, transient receptor potential cation channel, subfamily M, member 7; *MagT1*, magnesium transporter 1; *AQP1*, aquaporin 1; *AQP3*, aquaporin 3; *AQP7*, aquaporin 7; *AQP8*, aquaporin 8.

**Table 3 animals-15-01751-t003:** Effects of dietary PMS on serum levels of immune parameters in weaned piglets.

Item	PMS Supplementation, %						*p*-Value		
	0	0.15	0.30	0.45	0.60	0.75	Group	Linear	Quadratic
IgG (μg/mL)	0.25 ± 0.06	0.23 ± 0.09	0.28 ± 0.08	0.47 ± 0.11	0.40 ± 0.14	0.54 ± 0.13	0.14	0.01	0.03
IgM (mg/mL)	0.43 ± 0.02	0.39 ± 0.19	0.59 ± 0.08	0.67 ± 0.05	0.41 ± 0.11	0.59 ± 0.05	0.27	0.30	0.43
IgA (mg/mL)	11.56 ± 1.76	12.64 ± 3.17	17.10 ± 3.44	18.60 ± 3.66	14.45 ± 2.28	21.76 ± 3.25	0.09	0.10	0.28
IL-1β (ng/mL)	0.53 ± 0.09	0.42 ± 0.10	0.38 ± 0.11	0.32 ± 0.06	0.40 ± 0.04	0.08 ± 0.07	0.38	0.04	0.12
IL-2 (pg/mL)	20.09 ± 3.28	18.82 ± 3.18	19.35 ± 4.75	18.82 ± 3.03	23.33 ± 3.14	31.45 ± 4.98	0.09	0.01	0.02
IL-8 (pg/mL)	0.96 ± 0.13	0.74 ± 0.24	1.08 ± 0.28	1.06 ± 0.18	0.76 ± 0.24	1.10 ± 0.21	0.72	0.68	0.92
IL-10 (pg/mL)	14.90 ± 1.70 ^a^	9.64 ± 0.53 ^b^	9.24 ± 0.48 ^b^	9.44 ± 2.05 ^b^	15.28 ± 2.37 ^a^	15.61 ± 1.59 ^a^	0.03	0.28	0.01
TNF-α (ng/mL)	0.14 ± 0.04	0.10 ± 0.04	0.13 ± 0.04	0.11 ± 0.20	0.13 ± 0.01	0.11 ± 0.3	0.48	0.28	0.56

^a,b^ Means in a row without a common superscript letter differ significantly (*p* < 0.05). Data are presented as the mean + SE (*n* = 6). Abbreviations: IgG, Immunoglobulin G; IgM, Immunoglobulin M; IgA, Immunoglobulin A; IL-1β, Interleukin-1β; IL-2, Interleukin-2; IL-8, Interleukin-8; IL-10, Interleukin-10; TNF-α, Tumor inflammatory factor-α.

## Data Availability

Data from this study can be obtained upon request from the corresponding author. They are not publicly accessible due to privacy limitations.

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
