# Peer review of "Effect of Potassium–Magnesium Sulfate on Intestinal Dissociation and Absorption Rate, Immune Function, and Expression of NLRP3 Inflammasome, Aquaporins and Ion Channels in Weaned Piglets"

_animals, 2025, doi:10.3390/ani15121751_

Round 1

Reviewer 1 Report

Comments and Suggestions for Authors

The authors investigated the effects of PMS on intestinal dissociation and absorption rate, immune function, and expression of NLRP3 inflammasome, aquaporins and ion channels in weaned piglets from both in vitro and in vivo experiments. The study is interesting and the results of this study are important for the future application of PMS as new feed additive to mitigate weaning stress in piglets. However, minor revisions are necessary and detailed suggestions are as follows.

  1. How were the doses (0%、15%、0.30%、0.45%、0.60%、0.75%) of PMS selected for different treatments?
  2. What gender of the piglets used in this study?
  3. The author determined the ion contents in the feces, but the information of fecal collection was missing, which should be added in the Materials and Methods.
  4. Did the author determine the effect of PMS on intestinal barrier function of weaned piglets?
  5. Aquaporins have at least 13 subtypes, why choosing AQP1, 3, 7, and 8 for determinations? How about other subtypes?
  6. The replicate number should be provided in each table and figures.
  7. The weakness of this study should be added in the discussion. The authors only determined the mRNA expression of intestinal genes, and lacked of data concerning the potential molecular mechanism of PMS on these gene expression.
  8. More references in recent years about the influence of Mg or K on NLRP3 as well as the role of NLRP3 in piglets should be provided.
  9. The conclusion in the abstract should keep consistent with the Conclusion part.
  10. It seems that the effects of PMS on the expression of the same gene (such as TRMP6) varied in different intestinal segments? Please explain the possible reasons.

Author Response

Dear professor,

Thanks very much for the helpful comments. We have revised the manuscript accordingly as kindly suggested. All the revisions have been highlighted yellow in our revised manuscript (R1 version) for your quick reference. The detailed responses (in Red) to your kind comments are as follows. We hope that our revisions and responses have fully addressed all the concerns.

The responses to Reviewer 1’s comments

Comments 1: How were the doses (0%, 15%, 0.30%, 0.45%, 0.60%, 0.75%) of PMS selected for different treatments?

Responses 1: Thank you for pointing this out. The doses of PMS were selected based on our preliminary effectiveness evaluation assay of PMS and the recommended dose by the manufacturer of PMS.

Comments 2: What gender of the piglets used in this study?

Responses 2: The piglets used in this study consisted of half castrated males and half females in each replicate of different treatments.

Comments 3: The author determined the ion contents in the feces, but the information of fecal collection was missing, which should be added in the Materials and Methods.

Responses 3: Thanks for the kind suggestion. We have added more details about fecal collection in Materials and Methods as kindly suggested.

Comments 4: Did the author determine the effect of PMS on intestinal barrier function of weaned piglets?

Responses 4: Yes, we have determined the effect of PMS on intestinal barrier function, mainly on the mRNA expression of tight junctions. However, there was no significant effects of PMS on the mRNA expression of tight junctions, and the results were not shown here in this manuscript.

Comments 5: Aquaporins have at least 13 subtypes, why choosing AQP1, 3, 7, and 8 for determinations? How about other subtypes?

Responses 5: We agree with this comment that at least 13 subtypes have been identified in mammals. The reasons why we chose AQP1, 3, 7, and 8 for determinations were mainly based on previous studies reporting that these potentially important subtypes in the gastrointestinal tract were closely associated with gut health. However, it remained unclear whether PMS treatment could also regulate other subtypes, which requires further investigations.

Comments 6: The replicate number should be provided in each table and figures.

Responses 6: Thank you for pointing this out. Now the replicate number has been added in the description of each table and figures.

Comments 7: The weakness of this study should be added in the discussion. The authors only determined the mRNA expression of intestinal genes, and lacked of data concerning the potential molecular mechanism of PMS on these gene expression.

Responses 7: Agree. We have added the weakness of our study in the discussion part as kindly suggested.

Comments 8: More references in recent years about the influence of Mg or K on NLRP3 as well as the role of NLRP3 in piglets should be provided.

Responses 8: Thanks for your comment. More new references related to NLRP3 regulation have been added as kindly suggested.

  1. Wang, P., Zhang, J., Tian, Y., Yu, B., He, J., Yu, J., Zheng, P. Weaning stress aggravates defense response and the vurden of protein metabolism in low-birth-weight piglets. Animals (Basel). 2025, 15(10):1369.
  2. Wang, H.H., Huang, C.R., Lin, H.C., Lin, H.A., Chen, Y.J., Tsai, K.J., Shih, C.T., Huang, K.Y., Ojcius, D.M., Tsai, M.H., Tseng, K.W., Chen, L.C. Magnesium-enriched deep-sea water inhibits NLRP3 inflammasome activation and dampens inflammation. Heliyon. 2024. 10(15):e35136.
  3. Immanuel, C.N., Teng, B., Dong, B.E., Gordon, E.M., Luellen, C., Lopez, B., Harding, J., Cormier, S.A., Fitzpatrick, E.A., Schwingshackl, A., Waters, C.M. Two-pore potassium channel TREK-1 (K2P2.1) regulates NLRP3 inflammasome activity in macrophages. Am. J. Physiol. Lung Cell Mol. Physiol. 2024. 326(3):L367-L376.

Comments 9: The conclusion in the abstract should keep consistent with the Conclusion part.

Responses 9: Thank you for pointing this out. Now revised to be consistent.

Comments 10: It seems that the effects of PMS on the expression of the same gene (such as TRMP6) varied in different intestinal segments? Please explain the possible reasons.

Responses 10: Thank you for pointing this out. More explanations have been added in the discussion part.

Reviewer 2 Report

Comments and Suggestions for Authors

Generally, this manuscript is well-prepared and the results illustrated that PMS supplementation could improve immune function, and modulate intestinal genes involved in water and ion homeostasis to control post-weaning diarrhea in piglets. However, I think there are some minor changes should be made. Following minor revisions, I believe this manuscript is suitable for publication and create interests in the scientific community. Specific comments on the manuscript are as follows: 

  1. Line 65: An extra reference number needs to be deleted.
  2. Line 116: “Therefore”can be changed to other words.
  3. Line 158, 159 and 160: The formula used needs to be supplemented with authoritative references.
  4. Line 171: the requirement for 7-11kg refers tothe first weight stage of 1-28d in this study, and the authors should add the information that the second stage should be estimated based on the later weight stage (28-42d).
  5. Line 226:“P<0.05 indicates a significant difference, while 0.05<P<0.10 suggests a trend.” Modify as “A p-value less than 0.05 indicates a statistically significant difference, while a p-value between 0.05 and 0.10 suggests a potential trend worth further investigation.”
  6. Line 273: “Table 2”is changed to “Table 3”.
  7. What’s the data analysis method utilized in Figure 1 and Figure 2? Please clarify.
  8. The homogeneity of variance test is a prerequisite for conducting one-way analysis of variance. Has the author conducted a homogeneity of variance test before conducting factor analysis of variance?
  9. Interestingly, theresults found that dietary PMS significantly reduced the expression of some potassium channel and magnesium channel. Please provide more discussion on this part.
  10. Line 508, delete the word “weaned”.

Author Response

Dear professor,

Thanks very much for the helpful comments. We have revised the manuscript accordingly as kindly suggested. All the revisions have been highlighted yellow in our revised manuscript (R1 version) for your quick reference. The detailed responses (in Red) to your kind comments are as follows. We hope that our revisions and responses have fully addressed all the concerns.

The responses to Reviewer 2’s comments

Comments 1: Line 65: An extra reference number needs to be deleted.

Responses 1: Thank you for pointing this out. Now extra reference number [1] has been deleted.

Comments 2: “Therefore”can be changed to other words.

Responses 2: Thanks very much. Now changed to “Hence”.

Comments 3: Line 158, 159 and 160: The formula used needs to be supplemented with authoritative references.

Responses 3: Thanks for pointing this out. Now the related reference has been added as kindly suggested.

  1. Clarke, L. L. A guide to Ussing chamber studies of mouse intestine. Am. J. Physiol. Gastrointest. Liver Physiol. 2009. 296(6): G1151-1166.

Comments 4: Line 171: the requirement for 7-11kg refers to the first weight stage of 1-28d in this study, and the authors should add the information that the second stage should be estimated based on the later weight stage (28-42d).

Responses 4: Thanks for the kind suggestion. We have added the corresponding information as kindly suggested.

Comments 5: Line 226:“P<0.05 indicates a significant difference, while 0.05<P<0.10 suggests a trend.” Modify as “A p-value less than 0.05 indicates a statistically significant difference, while a p-value between 0.05 and 0.10 suggests a potential trend worth further investigation.”

Responses 5: Thank you for pointing this out. We have modified this sentence as kindly suggested.

Comments 6: Line 273: “Table 2”is changed to “Table 3”.

Responses 6: Sorry for the typos. Now revised.

Comments 7: What’s the data analysis method utilized in Figure 1 and Figure 2? Please clarify.

Responses 7: Thank you for pointing this out. The data analysis for in vitro assay used student’s t-test for comparison of intestinal dissociation and absorption rate, including Figure 1 and Figure2. The related information has been added in the 2.2.5 Statistical Analysis part.

Comments 8: The homogeneity of variance test is a prerequisite for conducting one-way analysis of variance. Has the author conducted a homogeneity of variance test before conducting factor analysis of variance?

Responses 8: We agree with this comment. Yes, we have conducted a homogeneity of variance test before performing one-way analysis of variance. The related information has been added in the 2.2.5 Statistical Analysis part.

Comments 9: Interestingly, the results found that dietary PMS significantly reduced the expression of some potassium channel and magnesium channel. Please provide more discussion on this part.

Responses 9: Thanks for your comment. More references have been added on the discussion part as kindly suggested.

  1. Wang, J., Zeng, L., Tan, B., Li, G., Huang, B., Xiong, X., Li, F., Kong, X., Liu, G., Yin, Y. Developmental changes in intercellular junctions and Kv channels in the intestine of piglets during the suckling and post-weaning periods. J. Anim. Sci. Biotechnol. 2016, 7:4.
  2. Pietropaolo, G., Pugliese, D., Armuzzi, A., Guidi, L., Gasbarrini, A., Rapaccini, G.L., Wolf, F.I., Trapani, V. Magnesium absorption in intestinal cells: evidence of cross-talk between EGF and TRPM6 and novel implications for cetuximab therapy. Nutrients. 2020, 12(11):3277.

Comments 10: Line 508, delete the word “weaned”.

Responses 10: Thank you for pointing this out. Now deleted.

Reviewer 3 Report

Comments and Suggestions for Authors

The authors conducted the in vitro and in vivo studies to evaluate the effects of potassium-magnesium sulfate (PMS) on intestinal dissociation and absorption rate, immune function, and expression of NLRP3 inflammasome, aquaporins and ion channels in weaned piglets. The manuscript was well written, and the experimental design was sound. The observations in this manuscript could provide the tool to improve gut immune function and water and ion homeostasis to maintain fecal consistence in piglets post weaning. However, there are few issues should be addressed before considering as the publication.

  1. In the Abstracts section, the optimal dosage of dietary PMS should be indicated in the conclusion part.
  2. In the Statistical analysis section, the information was not enough, and the analysis method for dose-related effect should be also added.
  3. In table 1, the units of vitamins should be unified.
  4. In the description of tables and figures, “data were presented as mean + SE”, instead of “mean + SEM”.
  5. In the in vivo study, a total of 5 different dosages were used, while the optimal dosage was not demonstrated, thus the possible reasons should be provided in the discussion section.

Author Response

Dear professor,

Thanks very much for the helpful comments. We have revised the manuscript accordingly as kindly suggested. All the revisions have been highlighted yellow in our revised manuscript (R1 version) for your quick reference. The detailed responses (in Red) to your kind comments are as follows. We hope that our revisions and responses have fully addressed all the concerns.

The responses to Reviewer 3’s comments

Comments 1: In the Abstracts section, the optimal dosage of dietary PMS should be indicated in the conclusion part.

Responses 1: Thank you for pointing this out. Now revised.

Comments 2: In the Statistical analysis section, the information was not enough, and the analysis method for dose-related effect should be also added.

Responses 2: Thanks very much. More information about the statistical analysis especially for the dose-related effect has been added as kindly suggested.

Comments 3: In table 1, the units of vitamins should be unified.

Responses 3: Thanks for the kind suggestion. Now the units of vitamins have been unified to mg/kg except for VA, VD, VE, and VK.

Comments 4: In the description of tables and figures, “data were presented as mean + SE”, instead of “mean + SEM”.

Responses 4: Thank you for pointing this out. We have revised the related description of tables and figures as kindly suggested.

Comments 5: In the in vivo study, a total of 5 different dosages were used, while the optimal dosage was not demonstrated, thus the possible reasons should be provided in the discussion section.

Responses 5: Thank you for pointing this out. More discussion about the dosages of PMS has been added.

Reviewer 4 Report

Comments and Suggestions for Authors

Effect of Potassium-Magnesium Sulfate on Intestinal Dissociation and Absorption Rate, Immune Function, and Expression of NLRP3 Inflammasome, Aquaporins, and Ion Channels in Weaned Piglets

The article has scientific merit; however, some points need to be adjusted, such as the introduction, which should be more objective regarding the study. Some sections described there are unnecessary and could be removed, for example, lines 82-84. Concerning aquaporins and the NLRP3 inflammasome, these topics could be explored further in the text.

Regarding the materials and methods, particularly with reference to assay I (in vitro), I believe it could be better described with more detail about the methodology used. Another point would be to present the animals' initial average weight along with their standard deviations and sex.

Regarding Table 1, I believe the way it is presented may cause confusion for the reader, as the ingredients and nutrients are shown side by side, and it is not self-explanatory.

The fecal collection should also be described in more detail (how many animals? directly from the rectum?). Another aspect that requires a more detailed description is the statistical analysis: it does not present the model, does not state that regression will be performed, and does not mention the Spearman correlation.

The results are presented clearly; however, the discussion does not explain some results, for example, the expression of aquaporins and their relationship with cytokines.

The conclusion needs to be more specific regarding how much PMS should be added, since the use of regression was not well elucidated, with no presentation in the text of the regressions and their maximum and minimum points for each variable that showed a significant effect.

Comments on the Quality of English Language

Nothing to note regarding the English, but I suggest that it be reviewed.

Author Response

Dear professor,

Thanks very much for the helpful comments. We have revised the manuscript accordingly as kindly suggested. All the revisions have been highlighted yellow in our revised manuscript (R1 version) for your quick reference. The detailed responses (in Red) to your kind comments are as follows. We hope that our revisions and responses have fully addressed all the concerns.

The responses to Reviewer 4’s comments

Comments 1: The article has scientific merit; however, some points need to be adjusted, such as the introduction, which should be more objective regarding the study. Some sections described there are unnecessary and could be removed, for example, lines 82-84. Concerning aquaporins and the NLRP3 inflammasome, these topics could be explored further in the text.

Responses 1: Thank you for the helpful comments. We have deleted the related contents within previous Lines 82-84. Moreover, the introduction part has been modified with more references concerning AQPs and NLRP3 inflammasome added as kindly suggested.

  1. Gong, T., Yang, Y., Jin, T., Jiang, W., Zhou, R. Orchestration of NLRP3 inflammasome activation by ion fluxes. Trends Immunol. 2018. 39(5):393-406.
  2. Li, C., Chen, M., He, X., Ouyang, D. A mini-review on ion fluxes that regulate NLRP3 inflammasome activation. Acta Biochim. Biophys. Sin. (Shanghai). 2021, 53(2):131-139.
  3. Zhang, T., Zhao, J., Liu, T., Cheng, W., Wang, Y., Ding, S., Wang, R. A novel mechanism for NLRP3 inflammasome activation. Metabol. Open. 2022. 13:100166.
  4. Shen, C., Luo, Z., Yu, C., Wei, Y., Zhang, Z., Han, Y., Zhang, H., Zhang, J., Xu, W., Xu, J. Effect of microbe-derived antioxidants on intestinal oxidative stress, NLRP3 inflammasome, morphologic structure, and growth performance in weanling piglets. J. Food. Sci. 2025. 90(3):e70064.
  5. Chen, X., He, X., Du, X., Huang, Z., Jia, G., Zhao, H. Dihydromyricetin attenuates lipopolysaccharide-induced intestinal injury in weaned piglets by regulating oxidative stress and inhibiting NLRP3 inflammasome. J. Anim. Sci. 2025. 103:skaf114.
  6. Lu, Q., Wang, N., Wen, D., Guo, P., Liu, Y., Fu, S., Ye, C., Wu, Z., Qiu, Y. Baicalin attenuates lipopolysaccharide-induced intestinal inflammatory injury via suppressing PARP1-mediated NF-κB and NLRP3 signalling pathway. Toxicon. 2024. 239:107612.
  7. Laforenza U. Water channel proteins in the gastrointestinal tract. Mol. Aspects Med. 2012. 33(5-6):642-50.
  8. Ikarashi, N., Kon, R., Sugiyama, K. Aquaporins in the colon as a new therapeutic target in diarrhea and constipation. Int. J. Mol. Sci. 2016. 17(7):1172.
  9. Camilleri M, Carlson P, Chedid V, Vijayvargiya P, Burton D, Busciglio I. Aquaporin expression in colonic mucosal biopsies from irritable bowel syndrome with diarrhea. Clin. Transl. Gastroenterol. 2019. 10(4):e00019.
  10. Lv H, Li Y, Xue C, Dong N, Bi C, Shan A. Aquaporin: targets for dietary nutrients to regulate intestinal health. J. Anim. Physiol. Anim. Nutr (Berl). 2022. 106(1):167-180.
  11. Gao, J., Yin, J., Xu, K., Han, H., Liu, Z., Wang, C., Li, T., Yin, Y. Protein level and infantile diarrhea in a postweaning piglet model. Mediators Inflamm. 2020. 2020:1937387.

Comments 2: Regarding the materials and methods, particularly with reference to assay I (in vitro), I believe it could be better described with more detail about the methodology used. Another point would be to present the animals' initial average weight along with their standard deviations and sex.

Responses 2: Thanks very much. The references of the in vitro assays have been added. More details about the materials and methods have been modified as kindly suggested. The initial body weight of piglets used in this study was 7.52 ± 0.02 kg. Each group had 6 replicates of 6 piglets (3 castrated males and 3 females) per replicate.

  1. Xu, W., Yang, Y. Drug sorption onto and release from soy protein fibers. J. Mater. Sci. Mater. Med. 2009. 20(12):2477-2486.
  2. Clarke, L. L. A guide to Ussing chamber studies of mouse intestine. Am. J. Physiol. Gastrointest. Liver Physiol. 2009. 296(6): G1151-1166.

Comments 3: Regarding Table 1, I believe the way it is presented may cause confusion for the reader, as the ingredients and nutrients are shown side by side, and it is not self-explanatory.

Responses 3: Thanks for the kind suggestion. Now Table 1 has been modified with the nutrient level shown at the bottom of the table, corresponding to the same stage of the ingredients.

Comments 4: The fecal collection should also be described in more detail (how many animals? directly from the rectum?). Another aspect that requires a more detailed description is the statistical analysis: it does not present the model, does not state that regression will be performed, and does not mention the Spearman correlation.

Responses 4: Thank you for pointing this out. The details of fecal collection have been added. The fecal samples involved each piglet (216) used in this study. Specially, on the last three days of the experiment (days 40-42), fecal samples of approximately 20-30 g per piglet were collected from each replicate (n = 6) continuously over a period of three days after feeding. The collected fecal samples within each replicate from three days were mixed together followed by stored in sealed bags at -20℃ until analysis for K+ and Mg2+ contents in the feces. Moreover, the descriptions of statistical analysis including the regression and Spearman correlation have been modified as kindly suggested.

Comments 5: The results are presented clearly; however, the discussion does not explain some results, for example, the expression of aquaporins and their relationship with cytokines.

Responses 5: Thank you for pointing this out. The potential relationship between AQPs and cytokines has been added in the discussion part.

  1. Rabolli, V., Wallemme, L., Lo Re, S., Uwambayinema, F., Palmai-Pallag, M., Thomassen, L., Tyteca, D., Octave, J.N., Marbaix, E., Lison, D., Devuyst, O., Huaux, F. Critical role of aquaporins in interleukin 1β (IL-1β)-induced inflammation. J. Biol. Chem. 2014. 289(20):13937-47.
  2. Kalita, A., Das, M. Aquaporins (AQPs) as a marker in the physiology of inflammation and its interaction studies with garcinol. Inflammopharmacology. 2024. 32(2):1575-1592.
  3. Lin, G., Lin, L., Chen, X., Chen, L., Yang, J., Chen, Y., Qian, D., Zeng, Y., Xu, Y. PPAR-γ/NF-kB/AQP3 axis in M2 macrophage orchestrates lung adenocarcinoma progression by upregulating IL-6. Cell Death Dis. 2024. 15(7):532.

Comments 6: The conclusion needs to be more specific regarding how much PMS should be added, since the use of regression was not well elucidated, with no presentation in the text of the regressions and their maximum and minimum points for each variable that showed a significant effect.

Responses 6: Thank you for pointing this out. We have modified the conclusion with specific recommendation level of PMS supplementation. The recommended doses of PMS supplemented into the basal corn-soybean basal diet would be 0.3%, based on the cost-effectiveness considerations of PMS and the findings from our current and previous studies (Cao et al., 2022).